# CUSTOMNET: ZERO-SHOT OBJECT CUSTOMIZATION WITH VARIABLE-VIEWPOINTS IN TEXT-TO-IMAGE DIFFUSION MODELS

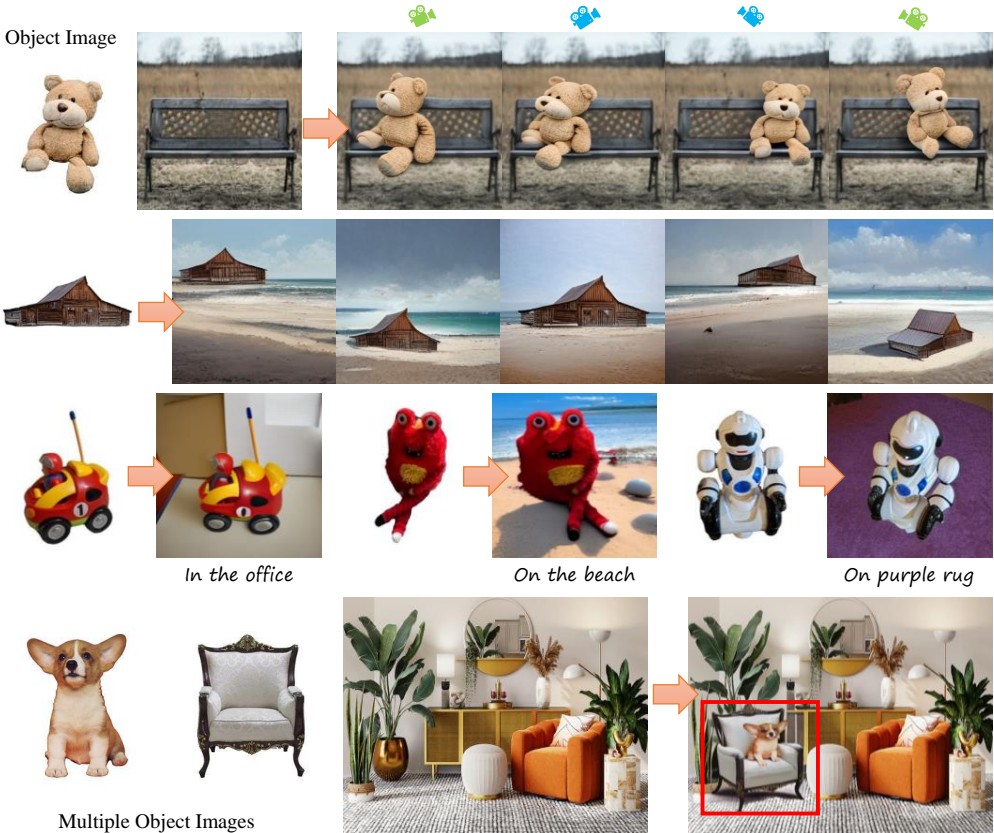

Figure 1: We proposed **CustomNet**, a zero-shot customization method that can generate harmonious customized images with explicit viewpoint, location, and background controls simultaneously, while ensuring identity preservation.

## ABSTRACT

Incorporating a customized object into image generation presents an attractive feature in text-to-image generation. However, existing optimization-based and encoder-based methods are hindered by drawbacks such as time-consuming optimization, insufficient identity preservation, and a prevalent copy-pasting effect. To overcome these limitations, we introduce CustomNet, a novel object customization approach that explicitly incorporates 3D novel view synthesis capabilities into the object customization process. This integration facilitates the adjustment of spatial position relationships and viewpoints, yielding diverse outputs while effectively preserving object identity. Moreover, we introduce intricate designs to enable location control and flexible background control through textual descriptions or specific user-defined images, overcoming the limitations of existing 3D novel view synthesis methods. We further leverage a dataset construction pipeline that can better handle real-world objects and complex backgrounds. Equipped with these designs, our method facilitates zero-shot object customization with-

out test-time optimization, offering simultaneous control over the location, viewpoints, and background. As a result, our CustomNet ensures enhanced identity preservation and generates diverse, harmonious outputs.

# 1 INTRODUCTION

Recently, there has been an emerging trend of the diffusion model to become the new state-of-the-art model in text-to-image (T2I) generation (Nichol et al., 2021; Ramesh et al., 2022; Saharia et al., 2022; Rombach et al., 2022). The society also applies extra diverse control conditions to the T2I diffusion models (Zhang & Agrawala, 2023; Mou et al., 2023; Li et al., 2023d), such as layout, style, and depth. Customization, as another control dimension in diffusion models, has received significant attention. It allows users to incorporate objects from reference images into the generated images while preserving their identities. Pioneering works such as Dreambooth (Ruiz et al., 2023) and Textual Inversion (Gal et al., 2022) use a few images of the same object to finetune the diffusion model parameters or learn concept word embeddings through an iterative optimization process. Although these optimization-based techniques excel at maintaining object identity, they suffer from certain drawbacks, such as time-consuming optimization and a tendency to overfit when only a single image is provided.

Consequently, researchers have started exploring encoder-based methods (Yang et al., 2023; Li et al., 2023d;a; Wei et al., 2023; Chen et al., 2023b). These methods only require training an encoder to explicitly represent visual concepts of objects. Once trained, the concept embeddings obtained by encoding the image can be directly fed into the denoising process during inference, achieving a speed comparable to the standard diffusion model sampling process. However, simply injecting an image into a compressed concept embedding often leads to inadequate identity preservation (Yang et al., 2023; Li et al., 2023a). To address this issue, several methods have been proposed to enhance detail preservation by introducing local features (Wei et al., 2023; Ma et al., 2023) or spatial details (Chen et al., 2023b). Despite these improvements, a closer examination of the results produced by these methods reveals a prevalent copy-pasting effect, *i.e.*, the objects in the synthesized image and source image are identical. The only variations observed stem from basic data augmentations applied during training, such as flipping and rotation. This limitation makes it difficult to achieve harmonized results with the background and negatively impacts output diversity.

When generating images with customized objects, it is necessary to consider the spatial position and viewpoint of the object in relation to the scene, *i.e.*, the 3D properties of the object, in order to achieve harmonious and diverse results. Following this guidance, we introduce *CustomNet*, a novel object customization method that facilitates diverse viewpoints in text-to-image diffusion models. Unlike previous encoder-based methods that either fail to maintain a strong object identity or suffer from apparent copy-pasting effects, CustomNet explicitly controls the viewpoint of the customized object. This capability results in diverse viewpoint outputs while effectively preserving identity. The primary advantage of CustomeNet stems from its utilization of 3D novel view synthesis, which allows for the prediction of outputs from other views based on a single image input. A representative work on 3D novel view synthesis is Zero-1-to-3 (Liu et al., 2023a), which employs the massive synthetic 3D object dataset Objaverse (Deitke et al., 2023) with multiple views to train a viewpoint-conditioned diffusion model featuring explicit viewpoint control. However, simply incorporating Zero-1-to-3 into object customization tasks poses several challenges due to its inherent limitations: 1) It is solely capable of generating centrally positioned objects, lacking the ability to place them in alternative locations; 2) It is unable to generate diverse backgrounds, being restricted to a simplistic white background. It also lacks the text control functionality to produce desired backgrounds. These constraints significantly hinder its applicability in object customization tasks.

In this paper, we make delicate designs to incorporate 3D novel view synthesis capability for object customization while adhering to the requirements of the customization. Our proposed CustomNet first integrates viewpoint control ability and subsequently supports location control to place objects in user-defined positions and sizes. The location control is achieved by concatenating the transformed reference object image to the UNet input. For background generation, we introduce a dual cross-attention module that enables CustomNet to accept both text (for background generation) and object images (for foreground generation). CustomNet also accommodates user-provided background images, generating harmonious results. Moreover, we have designed a dataset construction pipeline that effectively utilizes synthetic multiview data and massive natural images to better han-

dle real-world objects and complex backgrounds. Built upon those designs, CustomNet supports fine-grained object and background control within a unified framework and can achieve zero-shot customization with excellent identity preservation and diverse outcomes, as illustrated in Fig. 1.

We summarize our contributions as follows: **(1)**. In contrast to previous customization approaches that predominantly rely on 2D input images, we propose CustomNet to explicitly incorporate 3D novel view synthesis capabilities (*e.g.*, Zero-1-to-3) into the object customization process. This allows for the adjustment of spatial position relationships and viewpoints, leading to improved identity preservation and diverse outputs. **(2)**. Our approach features intricate designs that enable location control and flexible background control, addressing inherent limitations in Zero-1-to-3, such as simplistic white backgrounds, exclusively centered objects, and overly synthetic effects. **(3)**. We design a dataset construction pipeline that effectively utilizes synthetic multiview data and massive natural images to better handle real-world objects and complex backgrounds. **(4)**. Equipped with the aforementioned designs, our method enables zero-shot object customization without test-time optimization while controlling location, viewpoints, and background simultaneously. This results in enhanced identity preservation and diverse, harmonious outputs.

## 2 RELATED WORKS

**Object customization in diffusion models**. With the promising progress of text-to-image diffusion models (Sohl-Dickstein et al., 2015; Ho et al., 2020; Song & Ermon, 2019; Nichol et al., 2021; Ramesh et al., 2022; Saharia et al., 2022; Rombach et al., 2022), researches explore to capture the information of a reference object image and maintain its identity throughout the diffusion model generation process, *i.e.*, object customization. These methods can be broadly classified into optimization-based techniques Ruiz et al. (2023); Gal et al. (2022); Chen et al. (2023a); Liu et al. (2023b) and encoder-based approaches Yang et al. (2023); Song et al. (2023); Li et al. (2023d;a); Wei et al. (2023). Optimization-based methods can achieve high-fidelity identity preservation; however, they are time-consuming and may sometimes result in overfitting. In contrast, current encoder-based methods enable zero-shot performance but may either lose the identity or produce trivial results resembling copy-pasting. In contrast, our proposed CustomNet aims to preserve high fidelity while supporting controllable viewpoint variations, thereby achieving more diverse outcomes.

**Image harmonization**. In image composition, a foreground object is typically integrated into a given background image to achieve harmonized results. Various image harmonization methods Sunkavalli et al. (2010); Chen & Kae (2019); Cong et al. (2020); Guo et al. (2021) have been proposed to further refine the foreground region, ensuring more plausible lighting and color adjustments Xue et al. (2022); Cong et al. (2022); Chen et al. (2022). However, these methods focus on low-level modifications and are unable to alter the viewpoint or pose of the foreground objects. In contrast, our proposed CustomNet not only achieves flexible background generation using user-provided images but also offers additional viewpoint control and enhanced harmonization.

**3D novel view synthesis** aims to infer the appearance of a scene from novel viewpoints based on one or a set of images of a given 3D scene. Previous methods have typically relied on classical techniques such as interpolation or disparity estimation (Park et al., 2017; Zhou et al., 2018), as well as generative models (Sun et al., 2018; Chan et al., 2022). More recently, approaches based on Scene Representation Networks (SRN) (Sitzmann et al., 2019) and Neural Radiance Fields (NeRF) (Mildenhall et al., 2020; Yu et al., 2021; Jang & Agapito, 2021) have been explored. Furthermore, diffusion models have been introduced into novel view synthesis (Liu et al., 2023a; Watson et al., 2022). Zero-1-to-3 Liu et al. (2023a) propose a viewpoint-conditioned diffusion model trained on large synthetic datasets, achieving excellent performance in single-view 3D reconstruction and novel view synthesis tasks. Our CustomNet leverages the powerful 3D capabilities of diffusion models for object customization tasks, generating outputs with diverse viewpoints while preserving the identity

## 3 PROPOSED METHOD

**Overview.** Given a reference background-free [1] object image $x \in \mathbb{R}^{H \times W \times 3}$ with height $H$ and width $W$, we aim to generate a customized image $\hat{x}$ where the object of the same identity can

---

[1]Background-free images can be easily obtained by segmentation methods, *e.g.*, SAM (Kirillov et al., 2023).

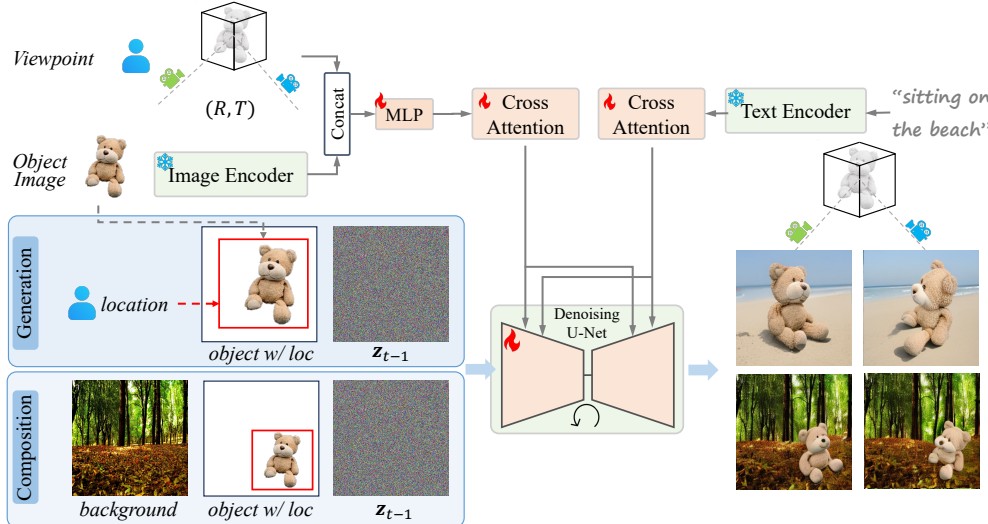

Figure 2: Overview of our proposed CustomNet. CustomNet is able to simultaneously control viewpoint, location, and background in a unified framework, thereby achieving harmonious customized image generation while effectively preserving object identity and texture details. The background generation can be controlled either through textual descriptions (the 'Generation' branch) or by providing a specific user-defined image (the 'Composition' branch).

be seamlessly placed in a desired environment (*i.e.*, background) harmoniously with appropriate viewpoint and location variations.

As illustrated in Fig. 2, we propose CustomNet, a novel architecture designed to achieve this customization conditioned on the viewpoint $[R, T]$ (where $R$ and $T$ represent the relative camera rotation and translation of the desired viewpoint, respectively), object location $L$, and background condition $B$. The background can be controlled either through textual descriptions or by providing a specific user-defined image $x_{bg}$. :

$$\hat{x} = \text{CUSTOMNET}(x, [R, T], L, B). \tag{1}$$

Our work is the first attempt to concurrently control viewpoint, location, and background in the customized synthesis task, thereby achieving a harmonious customized image while effectively preserving object identity and texture details (see Fig. 1 and Fig. 3 for representative results). Specifically, we observe that explicit viewpoint control is the missing ingredient for customization that enables simultaneous viewpoint alteration and object identity preservation (more details are in Sec. 4.3). Guided by this insight, our proposed CustomNet builds on the existing powerful viewpoint-conditioned diffusion model, Zero-1-to-3 (Liu et al., 2023a), which facilitates accurate viewpoint manipulation of the reference object. We further extend the location control by concatenating the reference object image transformed by the desired location and size into the UNet input. Moreover, we offer flexible background control through a textual description or a specific background image. This leads to improved identity preservation and diverse, harmonious outputs.

**Discussions.** There also exist simplistic approaches to achieving customization, which involve placing the object into a specific background while accounting for viewpoint and location variations in two distinct stages. Firstly, one can synthesize an object from the desired viewpoint using an existing single image-based novel view synthesis method, such as Zero-1-to-3. Subsequently, the synthesized object can be placed into a given background at the desired location using textual background inpainting (*e.g.*, SD-Inpainting model (RunwayML, 2022)) or exemplar image-based object inpainting (*e.g.*, Paint-by-Example Yang et al. (2023), AnyDoor Chen et al. (2023b)). Blended Latent Diffusion(Avrahami et al., 2023) uses text to inpaint a mask region of image. OVTrack(Li et al., 2023c) employs SD-inpainting for background with foreground objects in multi-object tracking. However, employing background inpainting often results in suboptimal harmonious outcomes with obvious artifacts. On the other hand, foreground object-based inpainting is prone to issues such as copying-and-pasting artifacts and identity loss, and faces challenges in handling view variations (see Sec. 4.2). Moreover, adopting a two-stage approach necessitates the utilization of distinct methods to accomplish the desired customization. This requires alternating between multiple tools for refining effects to enhance performance, thereby exacerbating the complexity of its usage. Instead, our CustomNet achieves the desired customization in a unified framework with more precise controls.

### 3.1 Control the Viewpoint and Location of Objects

**Viewpoint Control.** To enable synthesizing a target customized image complied with the given viewpoint parameter $[R, T]$, we follow the view-conditioned diffusion method introduced by Zero-1-to-3. As shown in Fig. 2 (the left-top part), we first apply a pre-trained CLIP Radford et al. (2021) image encoder to encode the reference background-free object image into an object embedding, containing high-level semantic information of the input object. Then the object embedding is concatenated with $[R, T]$ and passed through a trainable lightweight multi-layer perception (MLP). The fused object embedding further passes to the denoising UNet as a condition with the cross-attention mechanism to control viewpoints of the synthesized images.

**Location Control.** We further control the object location in the synthesized image by concatenating the reference object image with the desired location and size to the UNet input. The process is illustrated in Fig. 2 (the 'Generation' branch). The desired location and size $L$, represented as a bounding box $[x, y, w, h]$, are given by the user. Then, we resize the reference object image into the size of $[w, h]$ and place its left-top corner at the $[x, y]$ coordinate of a background-free image (this image is the same size as the target image being denoised but without background). The additional concatenated reference object image helps the model synthesize the desired image while keeping the identity and the texture details Rombach et al. (2022); Liu et al. (2023a). Note that Zero-1-to-3 directly concatenates the centrally-located reference object to the UNet input, which can only synthesize an image where the object is centered. Our method enables synthesizing the target object at the desired position with the proposed explicit location control.

### 3.2 Flexible Background Control by Text or Reference Images

Our proposed framework exhibits flexibility in generating backgrounds after positioning the object. There are two approaches. The first approach, referred to as *Generation*-based background control, involves synthesizing a target background using a textual description provided by the user, as shown in the 'Generation' branch in Fig. 2. The second approach, termed *Composition*-based background control, employs a specific background image supplied by the user (as illustrated in the 'Composition' branch in Fig. 2).

**Generation-based Background Control.** In this mode, the diffusion model takes $[\mathbf{z}_{t-1}, object\ w/\ loc]$ as inputs, where $\mathbf{z}_{t-1}$ represents the noisy latent at the time step $t-1$. The diffusion model is required to generate an appropriate background based on the textual description. Different from Zero-1-to-3, which solely accepts the object embedding without textual descriptions for background, we propose a novel dual cross-attention conditioning strategy that accepts both the fused object embedding with viewpoint control and textual descriptions for background. The dual cross-attention mechanism integrates the fused object embedding and the textual embedding through two distinct cross-attention modules. Specifically, we first employ the CLIP text encoder to obtain the textual embeddings and subsequently inject them into the denoising UNet, along with the fused object embedding, using the DUALATTN:

$$\text{DUALATTN}(Q, K_o, V_o, K_b, V_b) = \text{Softmax}(\frac{QK_o^T}{\sqrt{d}})V_o + \text{Softmax}(\frac{QK_b^T}{\sqrt{d}})V_b, \qquad (2)$$

where the query features $Q$ come from the UNet, while $K_o, V_o$ are the object features projected from fused object embeddings with viewpoint control, and $K_b, V_b$ are the background features from the textural embeddings. $d$ is the dimension of the aforementioned feature embeddings. During training, we randomly drop the background text description to disentangle the viewpoint control and background control. This straightforward yet effective design enables us to achieve accurate background control without affecting the object viewpoint control.

**Composition-based Background Control.** In many practical scenarios, users desire to seamlessly insert objects into pre-existing background images with specific viewpoints and locations. Our proposed CustomNet is designed to accommodate this functionality. More precisely, we extend the input channels of the UNet by concatenating the provided background image channel-wise, adhering to the Stable Diffusion inpainting pipeline. Consequently, the diffusion model accepts $[\mathbf{z}_{t-1}, object\ w/\ loc, x_{bg}]$ as inputs. Note that in this mode, the textual description is optional, allowing for straightforward input of NULL to the text prompt. In comparison to existing image composition methods (Yang et al., 2023; Chen et al., 2023b) which often struggle with copy-pasting artifacts or identity loss issues, our method offers viewpoint and location control over objects.

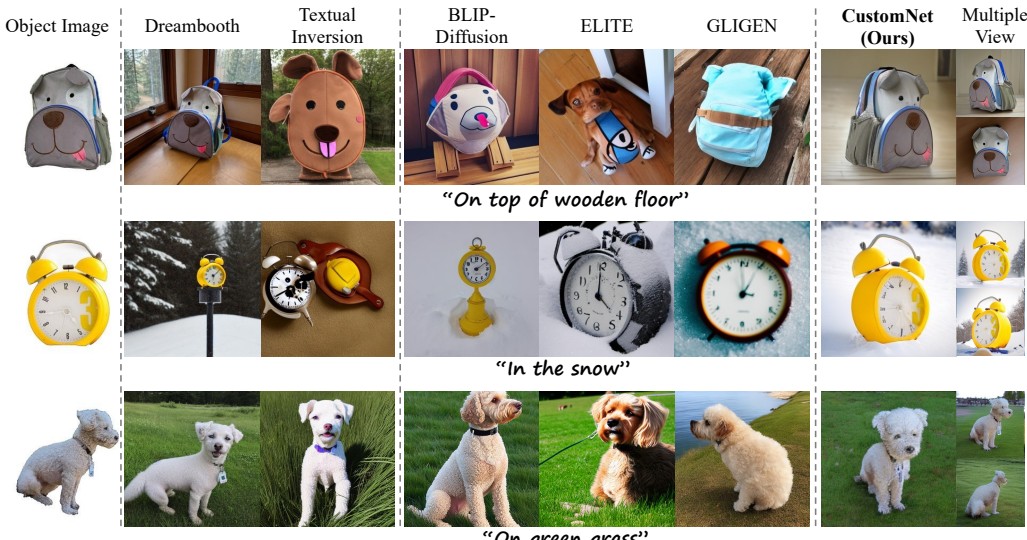

Figure 3: Qualitative comparison. Our CustomNet demonstrates superior capacities in terms of identity preservation, viewpoint control, and harmony of the customized image.

### 3.3 TRAINING STRATEGIES

**Data Construction Pipeline.** To train the proposed CustomNet, paired data with object image $x$, target image $x_{tgt}$, desired viewpoint parameter $[R, T]$, location and size $L$, and background condition $B$ (*i.e.*, a textual description or a background image) are required. Naturally, we can obtain multi-view object images and corresponding view parameters from existing 3D datasets like Objaverse Deitke et al. (2023). However, these datasets only contain object images without a background (usually with a pure black or white background), which is not appropriate for the customization task. As a result, we can simply perform mask-blending with the object image and collected background images. In addition, we use BLIP2 Li et al. (2023b) to caption the textual descriptions of the blended images for the background control with text prompts. However, since the composition between the object and background would be unreasonable (*i.e.*, the object is placed into the background disharmoniously) and the blended target image is unrealistic, the model trained on them often generates a disharmonious customized image, *e.g.*, the objects float over the background (see Sec. 4.3).

To alleviate this problem, we propose a training data construction pipeline that is the reverse of the above-mentioned way, *i.e.*, directly utilizing natural images as the target image and extracting objects from the image as the reference. Specifically, for a natural image, we first segment the foreground object using SAM model Kirillov et al. (2023). Then we synthesize a novel view of the object using Zero-1-to-3 with randomly sampled relative viewpoints. The textual description of the image can be also obtained using the BLIP2 model. In this way, we can synthesize a large amount of data pairs from natural image datasets, like OpenImages Kuznetsova et al. (2020). Meanwhile, the model trained with these data can synthesize more harmonious results with these natural images. More details are in the Appendix.

**Model Training.** Given paired images with their relative camera viewpoint, object locations, and background conditions (textual description or background image) $\{x, x_{tgt}, [R, T], L, B\}$, we can fine-tune a pre-trained diffusion model condition on these explicit controls. We adopt the viewpoint-conditioned diffusion model from Zero-1-to-3 (Liu et al., 2023a) as our base model, which also utilizes the latent diffusion model (LDM) (Rombach et al., 2022) architecture. LDM contains a variational auto-encoder with an encoder $\mathcal{E}$, decoder $\mathcal{D}$ and an UNet denoiser $\epsilon_\theta$. It performs the diffusion-denoising process in the latent space rather than the image space for efficiency. The optimization objective is:

$$\min_\theta \mathbb{E}_{z_{tgt}, \epsilon, t} \| \epsilon - \epsilon_\theta(z_{tgt}, t, c(x, [R, t], L, B)) \|^2, \tag{3}$$

where $z_{tgt} = \mathcal{E}(x_{tgt})$, and $c(\cdot)$ is the condition mechanism with explicit controls. Once the denoising UNet $\epsilon_\theta$ is trained, we can perform harmonious customization conditioned on the target viewpoint, location, and background with CustomNet.

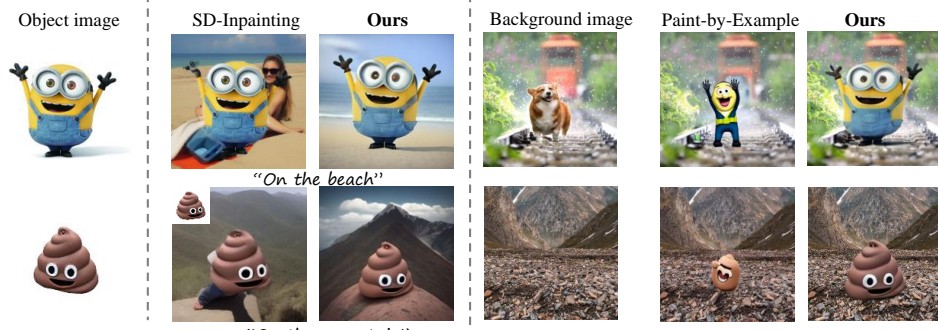

Figure 4: Comparison to existing textual background inpainting method SD-Inpainting model and foreground object inpainting model Paint-by-Example. Our CustomNet can achieve a more harmonious output with diverse viewpoint changes while preserving identity.

Table 1: **Quantitative Comparison.** We compute **DINO-I**, **CLIP-I**, **CLIP-T** following (Li et al., 2023a). We also conducted a user study to measure subjective metrics: **ID**, **View**, **Text** representing identity preservation, viewpoints variation, and text alignment, respectively.

| Method | DINO-I ↑ | CLIP-I ↑ | CLIP-T ↑ | ID ↑ | View ↑ | Text ↑ |
|---|---|---|---|---|---|---|
| DreamBooth (Ruiz et al., 2023) | 0.6333 | 0.8019 | 0.2276 | 0.1333 | 0.0822 | 0.1378 |
| Textual Inversion (Gal et al., 2022) | 0.5116 | 0.7557 | 0.2088 | 0.0100 | 0.0944 | 0.0367 |
| BLIP-Diffusion (Li et al., 2023a) | 0.6079 | 0.7928 | 0.2183 | 0.0522 | 0.0878 | 0.0422 |
| ELITE (Wei et al., 2023) | 0.5101 | 0.7675 | **0.2310** | 0.0056 | 0.0722 | 0.1056 |
| GLIGEN (Li et al., 2023d) | 0.5587 | 0.8152 | 0.1974 | 0.0111 | 0.0722 | 0.0144 |
| **CustomNet (Ours)** | **0.7742** | **0.8164** | 0.2258 | **0.7878** | **0.5912** | **0.6633** |

## 4 EXPERIMENTS

### 4.1 TRAINING DATASETS AND IMPLEMENTATION DETAILS

We use multi-view synthetic dataset Objaverse (Deitke et al., 2023), natural image dataset OpenImages-V6 (Kuznetsova et al., 2020) filtered as BLIP-Diffusion (Li et al., 2023a) to construct data pairs with the pipeline introduced in Sec. 3.3. A total of $(250+500)$K data pairs are constructed for model training. We exploit the Zero-1-to-3 checkpoint as the model weight initialization. For training, we employ AdamW (Loshchilov & Hutter, 2017) optimizer with a constant learning rate $2 \times 10^{-6}$ for 500K optimization steps. The total batch size is 96, and about 6 days are taken to finish the training on 8 NVIDIA-V100 GPUs with 32GB VRAM.

### 4.2 COMPARISON TO EXISTING METHODS

We compare our CustomNet to the optimization-based methods Textual Inversion (Gal et al., 2022), Dreambooth (Ruiz et al., 2023), and encoder-based (zero-shot) method GLIGEN Li et al. (2023d), ELITE Wei et al. (2023), BLIP-Diffusion Li et al. (2023a). We use their official implementation (for GLIDEN, ELITE, and BLIP-Diffusion) or the diffuser implementations von Platen et al. (2022) (for Textual Inversion, Dreambooth) to obtain the results. Note that Dreambooth requires several images of the same object to finetune.

Figure 3 shows the images generated with different methods (more results are in the Appendix). We see that the zero-shot methods GLIGEN, ELITE, BLIP-Diffusion, and the optimization-based method Textual Inversion are far from the identity consistent with the reference object. Dreambooth and the proposed CustomNet achieve highly promising harmonious customization results, while our method allows the user to control the object viewpoint easily and obatain diverse results. In addition, our method does not require time-consuming model fine-tuning and textual embedding optimization.

We also evaluate the synthesized results quantitatively. All methods apply 26 different prompts to perform customizations 3 times randomly on 50 objects. We calculate the visual similarity with CLIP image encoder and DINO encoder, denoted as **CLIP-I** and **DINO-I**, respectively. We measure the text-image similarity with CLIP directly, denoting **CLIP-T**. Tab. 1 shows the quantitative results, where CustomNet achieves better identity preservation (**DINO-I** and **CLIP-I** than other methods. Meanwhile, CustomNet shows comparable capacity to the state-of-the-art methods regarding textual control (**CLIP-T**). We also conducted a **user study** and collected 2700 answers for *Identity similarity* (ID), *View variation* (View), and *Text alignment* (Text), respectively. As shown in the right part of Tab. 1, most participants prefer CustomNet in all three aspects (78.78%, 64.67%, 67.84%).

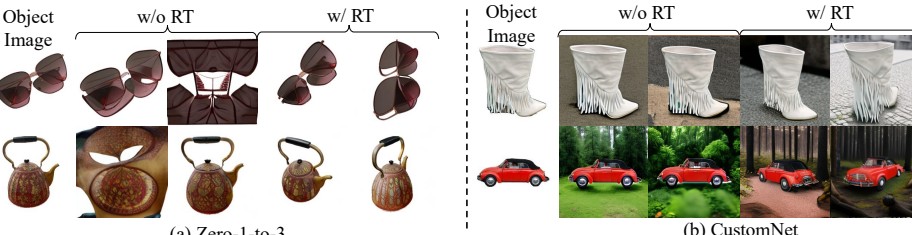

(a) Zero-1-to-3                    (b) CustomNet

Figure 5: Explicit viewpoints control. Without the explicit viewpoint parameters $[R, T]$, a) Zero-1-to-3 tends to generate images that cannot change the viewpoint or have undesired artifacts; b) CustomNet easily obtains copy-pasting effects, even though it is trained with the multi-view dataset.

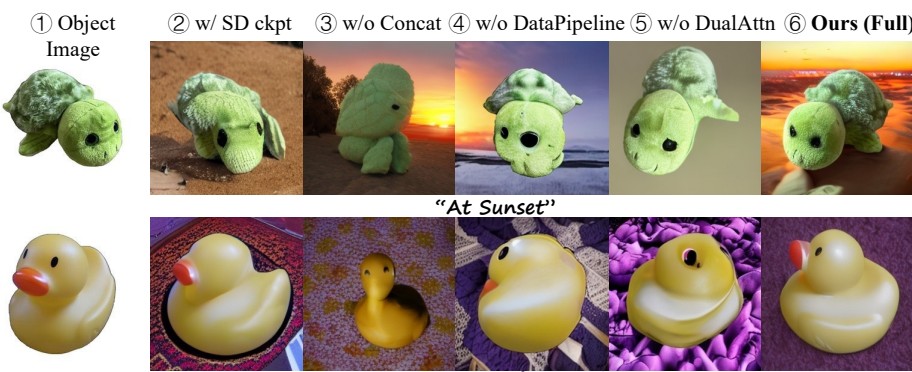

Figure 6: Ablation Study. *w/ SD ckpt*: initialize model weights with Stable-Diffusion pretrained checkpoints. *w/o Concat*: do not concatenate object with UNet input. *w/o DataPipeline*: do not use the dataset construction pipeline for OpenImages. *w/o DualAttn*: concatenate image and text embedding together and use shared cross-attn modules.

**Comparison to Inpainting-based Methods** Existing inpainting-based methods (SD-Inpainting model, Paint-by-Example Yang et al. (2023), AnyDoor Chen et al. (2023b)) can also place a reference object in the desired background in an inpainting pipeline. Given an object, the background can be inpainted with textual descriptions in the SD-Inpainting model, while this kind of method easily suffers from unreal and disharmonious results and cannot cast variations to the reference object. Our CustomNet can obtain more harmonious customization with diverse viewpoint control. Another line of methods Paint-by-Example and AnyDoor can inpaint the reference object to a given background image. AnyDoor has not open-sourced yet and we compare CustomNet with Paint-by-Example in Fig. 4. From Fig. 4, we see that Paint-by-Example cannot maintain the identity and differs significantly from the reference object.

## 4.3    ABLATION STUDIES AND ANALYSIS

We conduct detailed ablation studies to demonstrate the effectiveness of each design in CustomNet and the necessity of explicit control for identity preservation in harmonious customization.

***Explicit viewpoint control is the key for customization that enables simultaneous viewpoint alteration and object identity preservation.*** We conduct a comparison in terms of with and without explicit viewpoint control parameters $[R, T]$ on the original Zero-1-to-3 model. As shown in the left part of Fig. 5, models trained without viewpoint conditions tend to generate images that cannot change the viewpoint or have undesired artifacts. The same phenomenon can be observed in our CustomNet. To to specific, as shown in the right part of Fig. 5, without the explicit camera pose control, our model can only obtain copying-and-pasting effects, even though it is trained with the multi-view dataset. Note that in this setting, we also concatenate the object image into the UNet input, otherwise, the model cannot preserve adequate identity.

***Pretraining with massive multi-view data is important to preserve identity in CustomNet.*** We adopt Zero-1-to-3 as the initialization, *i.e.*, our CustomNet is pre-trained with massive multi-view Objarverse data, so that view-consistency information has been already encoded into the model. When we train CustomNet from the SD checkpoint (see the 2nd column in Fig. 6), the synthesized images cannot maintain view consistency from other viewpoints and suffer from quality degradation.

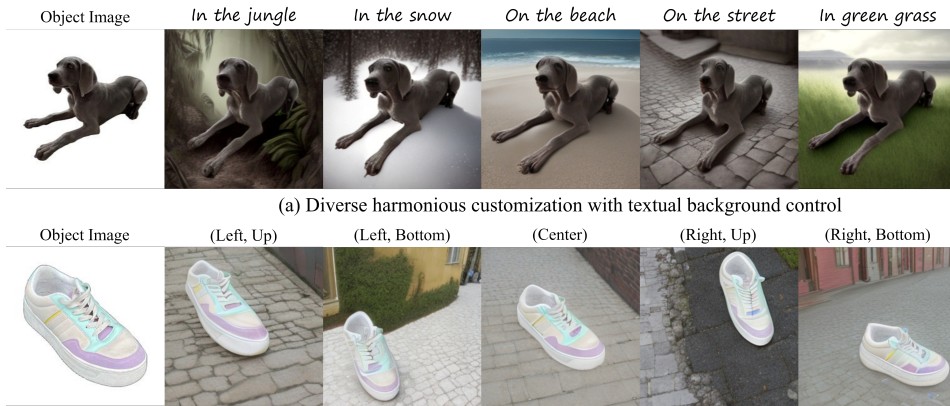

Object Image | In the jungle | In the snow | On the beach | On the street | In green grass

(a) Diverse harmonious customization with textual background control

Object Image | (Left, Up) | (Left, Bottom) | (Center) | (Right, Up) | (Right, Bottom)

(b) Precise location control

Figure 7: (a) Diverse background control with textual descriptions. (b) Precise location control with explicit location condition.

***Input concatenation helps better maintain texture details.*** Previous methods Wei et al. (2023); Chen et al. (2023b) also try to integrate local features to maintain texture details in synthesized images. In our method, concatenating the reference object to the UNet input can also preserve textures. Without the concatenation (see the 3rd column in Fig. 6), the color, shape, and texture of the generated images differ significantly from the reference object image. We also note that the model would generate copying-and-pasting images without any view variations when we do not adopt explicit viewpoint control (see Fig. 5). This is to say, the combination of input concatenation and explicit view conditions enables precise and harmonious customization.

***Our data construction pipeline enables more harmonious outputs.*** We adopted a new data construction pipeline for utilizing the OpenImages dataset in Sec. 3.3. Without this design, the model trained with only the data constructed by the naive combination between multi-view object images in Objaverse and background images can result in unrealistic and unnatural customized results, usually leading to 'floating' artifacts (see the 4th column in Fig. 5).

***Dual cross-attention enables disentangled object and background controls.*** We introduce dual-attention for the disentangled object-level control and background control with textual descriptions. When directly concatenating the text embedding and fused object embedding as the condition to be injected into the UNet, the model tends to learn a coupled control with viewpoint parameters and textual description. As a result, the viewpoint control capacity would degrade significantly and the model cannot generate desired background (see 5th column in Fig. 6).

### 4.4 MORE APPLICATIONS

**Diverse Background Control.** As shown in the first row of Fig. 7, our method can also generate harmonious results with diverse backgrounds controlled by textual descriptions. Thanks to the disentangled dual-attention design, the viewpoint of the object in the synthesized image can remain the same under different textual prompts.

**Precise Location Control.** With explicit location control design, our method also places an object into the desired location shown in the second row of Fig. 7, where the location is specified by the user and the background is synthesized directly by the textual description.

### 5 CONCLUSION

We present CustomNet, a novel object customization approach explicitly incorporating 3D view synthesis for enhanced identity preservation and viewpoint control. We introduce intricate designs for location control and flexible background control through textual descriptions or provided background images, and develop a dataset construction pipeline to handle real-world objects and complex backgrounds effectively. Our experiments show that CustomNet enables diverse zero-shot object customization while controlling location, viewpoints, and background simultaneously.

**Limitations**. CustomNet inherits Zero-1-to-3's resolution limitation ($256 \times 256$) restricting the generation quality. Despite outperforming existing methods, CustomNet cannot perform non-rigid transformations or change object styles. Future work aims to address these limitations.

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

## A  APPENDIX

### A.1  DATA CONSTRUCTION PIPELINE

We show our data construction pipeline in Fig. 8. For the 3D object dataset - Objarverse (Deitke et al., 2023), we can obtain multi-view object images and corresponding view parameters. However, these datasets only contain object images without a background (usually with a pure white background), which is not appropriate for the customization task. As a result, we can simply perform mask-blending with the object image and collected background images. However, since the composition between the object and background would be unreasonable (*i.e.*, the object is placed into the

background disharmoniously) and the blended target image is unrealistic, the model trained on them often generates a disharmonious customized image, *e.g.*, the objects float over the background.

We show our proposed data construction pipeline for natural images in Fig. 8 (b). It is the reverse of the first way, *i.e.*, directly utilizing natural images as the target image and extracting objects from the image as the reference. Specifically, for a natural image, we first use BLIP-2 5T (Li et al., 2023b) to extract the foreground object with the instruction {"image": image, "prompt": "Question: What foreground objects are in the image? find them and separate them using commas. Answer:"}. Then we feed the object and its corresponding text to SAM, SAM can receive text as input and output the segmentation mask of the corresponding object. Then we synthesize a novel view of the object using Zero-1-to-3 with randomly sampled relative viewpoints. In this way, we can synthesize a large amount of data pairs from natural image datasets, like OpenImages (Kuznetsova et al., 2020). Fig. 9 shows some data samples from the two dataset construction pipelines.

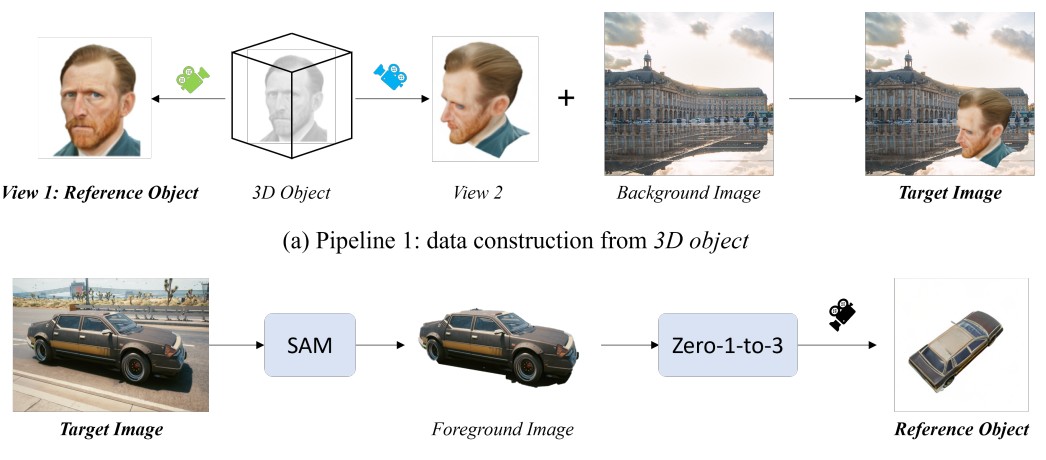

(a) Pipeline 1: data construction from *3D object*

(b) Pipeline 2: data construction from *single image*

Figure 8: Data construction pipeline from 3D objects (a) and single image (b).

## A.2 DEFINITION OF CAMERA MODEL

the definition and design of **RT** is consistent with the setting in Zero-1-to-3. In Zero-1-to-3, they render 3D objects to images with a predefined camera pose that always points at the center of the object. This means the viewpoint is actually controlled by polar angle $\theta$, azimuth angle $\phi$, and radius $r$ (distance away from the center), resulting all objects to be central of the rendered image. Therefore, the rotation matrix **R** can only control polar angle $\theta$, azimuth angle $\phi$, and the translation vector **T** can only control the radius $r$ between the object and camera. Thus the Zero-1-to-3 can only synthesize objects at the image center, lacking the capacity to place the object at the arbitrary location in the image. We tackle this problem by specifying the bounding box at the desired location to the latent.

## A.3 IMPROVEMENTS OF ZERO-1-TO-3

We show our improvements of Zero-1-to-3 compared with the limitations of Zero-1-to-3 in Fig. 10.

In the 1st, 2nd row, We compare with Zero-1-to-3 the ability to control the object location generation. As Zero-1-to-3 can only control polar angle $\theta$, azimuth angle $\phi$, and radius $r$, it can control the object location directly, so we apply our location control method on Zero-1-to-3. We resize the object and move it to the location in different bounding boxes (top left and top right in 1st row, bottom left and bottom right in 2nd row.). We can see that Zero-1-to-3 generate distortion in the generated images.

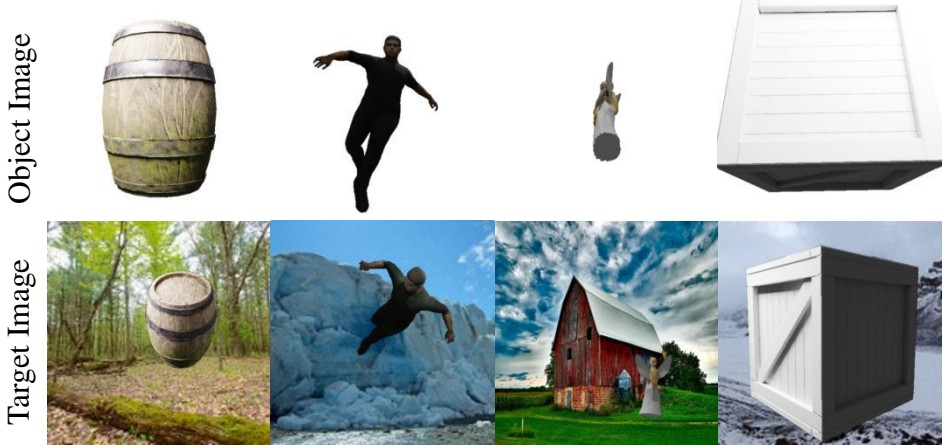

(a) Data samples from data construction pipeline 1.

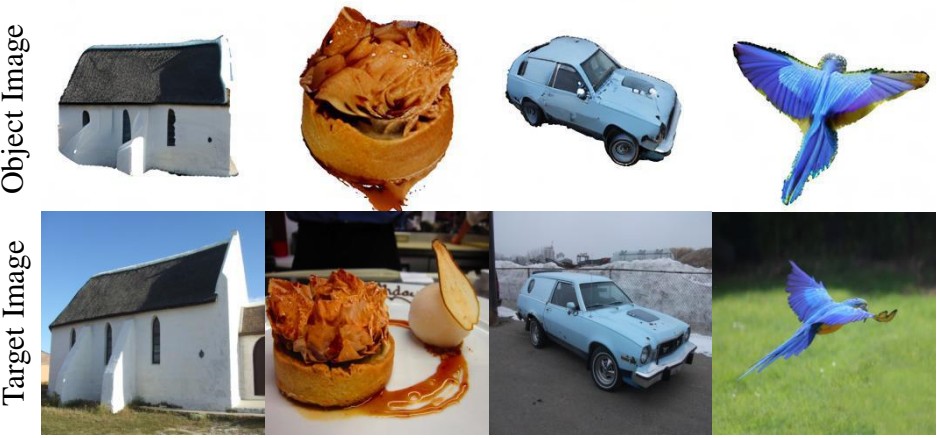

(b) Data samples from data construction pipeline 2.

Figure 9: Data samples from our constructed synthetic datasets and real world datasets. (a) shows synthetic data samples from data construction pipeline 1. (b) shows real world data samples from data construction pipeline 2.

This is because Zero-1-to-3 only trained on the object centred image, it can not really control the object location.

In the 3rd, 4th row, We compare with Zero-1-to-3 the ability of novel-view synthesis. We can see from the figure that Zero-1-to-3 fails to generate the resonable geometry for the dog and cartoon character. Our CustomNet, training with our real world constructed datasets, have a better comprehension of object geometry. We can generate a normal dog and a resonable cartoon chracter in multi-views.

In all rows, our CustomNet generate harmonious customized image with the control of different text prompts.

## A.4 MORE COMPARISON RESULTS

We provide additional qualitative comparison results to demonstrate the superiority of the proposed CustomNet shown in Fig. 11.

Object Image      Zero-1-to-3      Ours

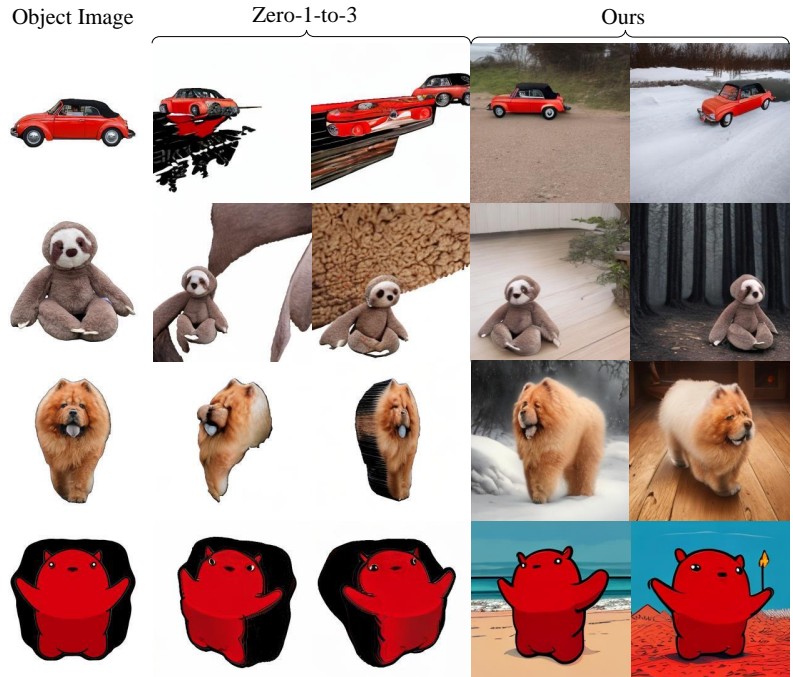

Figure 10: The comparison of CustomNet and Zero-1-to-3. In the 1st, 2nd row, We compare with Zero-1-to-3 the ability to control the object location generation. As Zero-1-to-3 can only control polar angle $\theta$, azimuth angle $\phi$, and radius $r$, it can control the object location directly, so we apply our location control method on Zero-1-to-3. We resize the object and move it to the location in different bounding boxes (top left and top right in 1st row, bottom left and bottom right in 2nd row.). We can see that Zero-1-to-3 generate distortion in the generated images. This is because Zero-1-to-3 only trained on the object centred image, it can not really control the object location.In the 3rd, 4th row, We compare with Zero-1-to-3 the ability of novel-view synthesis. We can see from the figure that Zero-1-to-3 fails to generate the resonable geometry for the dog and cartoon character. Our CustomNet, training with our real world constructed datasets, have a better comprehension of object geometry. We can generate a normal dog and a resonable cartoon chracter in multi-views. In all rows, our CustomNet generate harmonious customized image with the control of different text prompts.

## A.5 MORE RESULTS OF CUSTOMNET

We provide more real world object customized results shown in Fig. 12, Fig. 13 and 14.

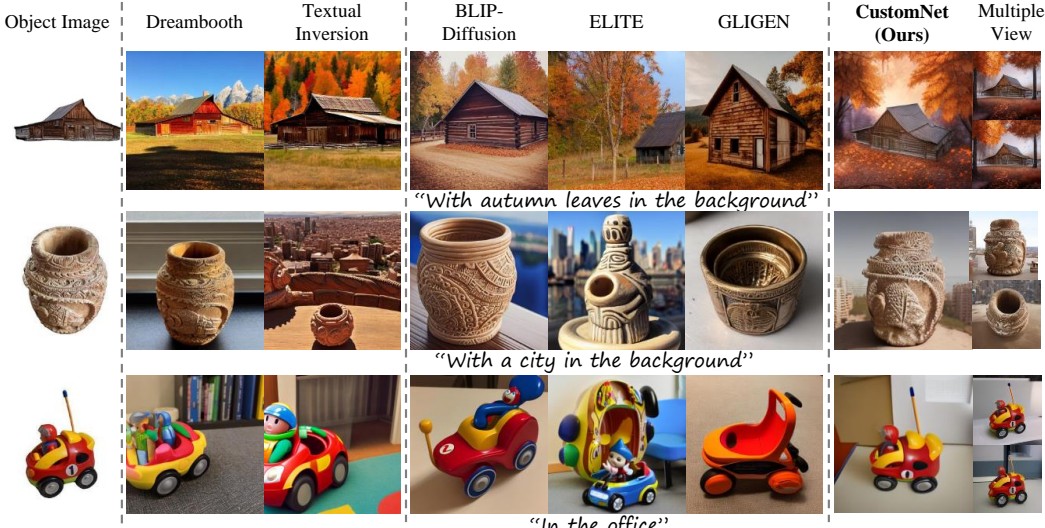

Figure 11: More qualitative comparison results. Our CustomNet demonstrates superior capacities in terms of identity preservation, viewpoint control, and harmony of the customized image.

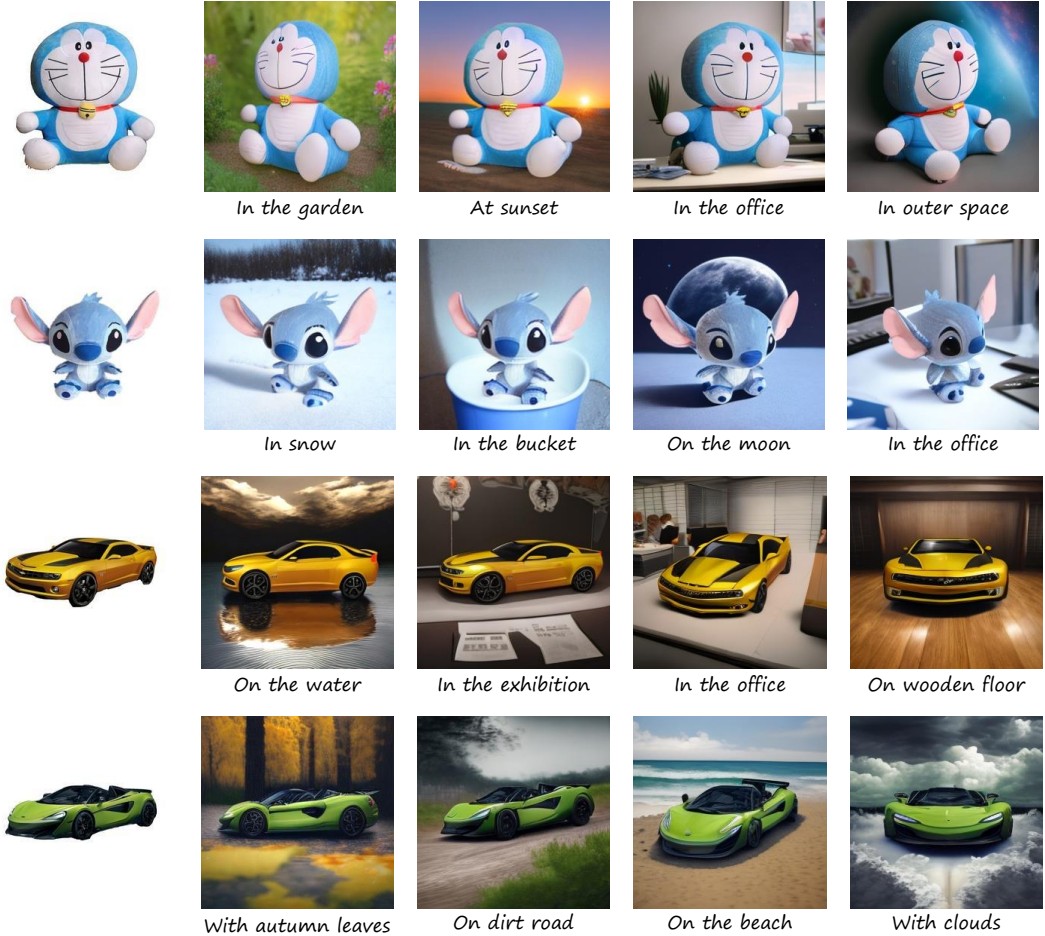

Figure 12: More real world object customized results of the proposed CustomNet.

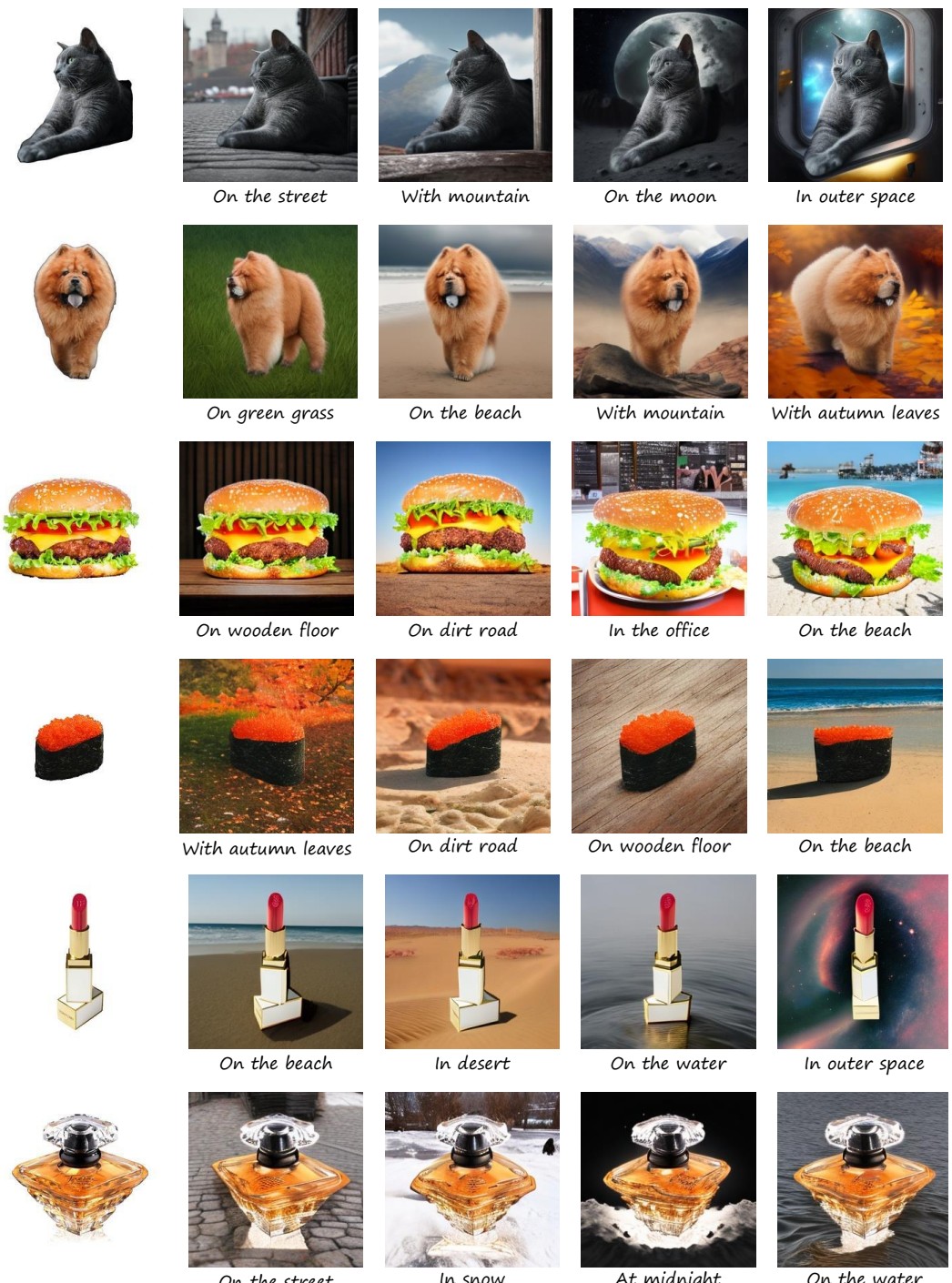

Figure 13: More real world object customized results of the proposed CustomNet.

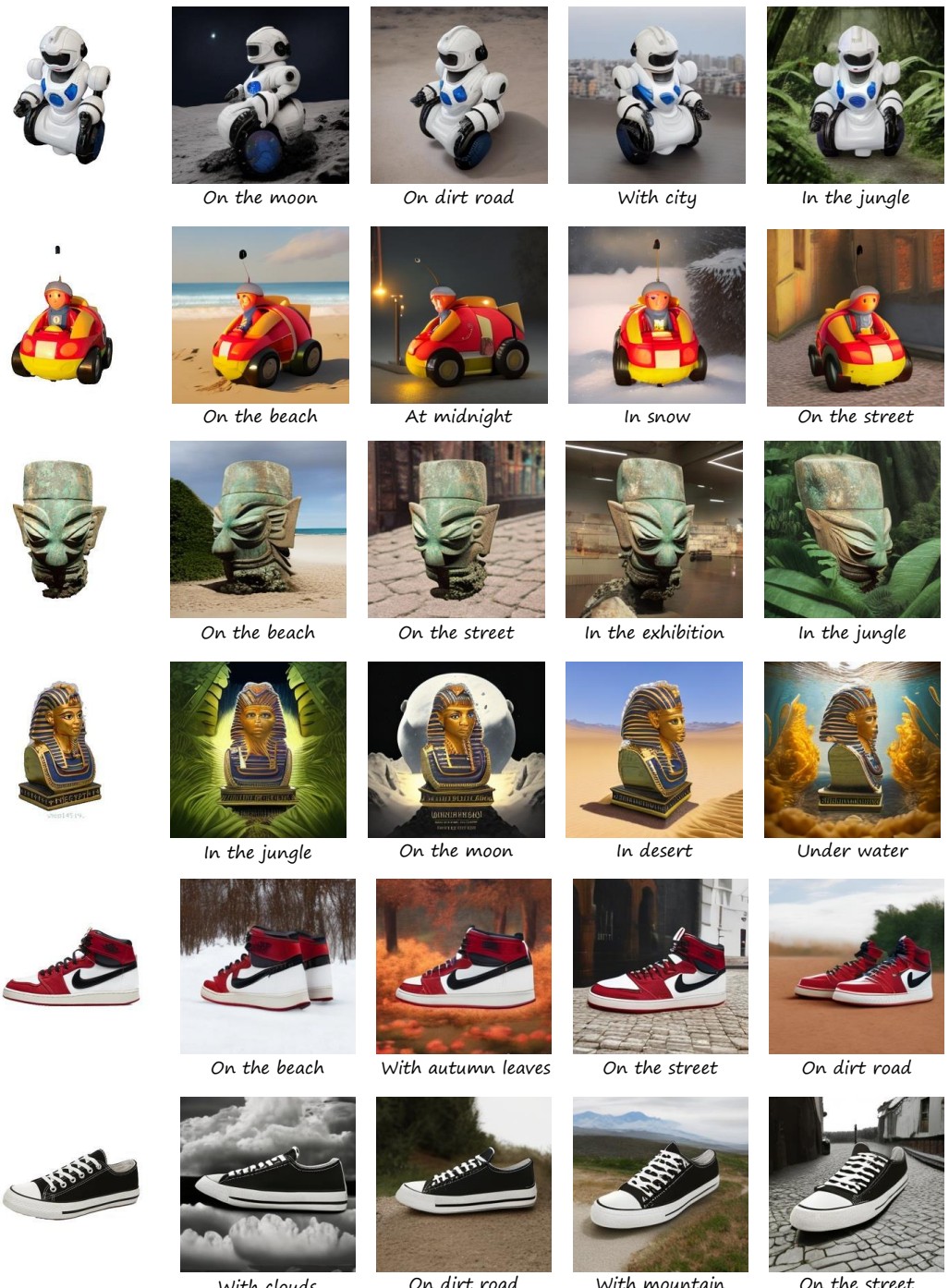

Figure 14: More real world object customized results of the proposed CustomNet.

