# OpenReview forum: "CustomNet: Zero-shot Object Customization with Variable-Viewpoints in Text-to-Image Diffusion Models"
_ICLR.cc/2024/Conference — Submitted to ICLR 2024_

### Official Review · Reviewer_h7FN · 2023-10-23

**Soundness:** 3 good
**Presentation:** 3 good
**Contribution:** 3 good
**Rating:** 6
**Confidence:** 4

**Summary:**

This paper proposes a CustomNet, an innovative approach to customizing objects in image generation based on SDs. It overcomes the limitations of existing methods, including slow optimization, identity preservation issues, and copy-pasting effects. It incorporates 3D novel view synthesis, allowing for diverse and identity-preserving outputs. It provides precise location and background control, surpassing existing techniques. This method enables zero-shot object customization without test-time optimization, offering better control and enhanced visual results.

**Strengths:**

1. Incorporating 3D novel view synthesis differentiates the proposed CustomNet from prior approaches, improving identity preservation and varied output generation.

2. This method's intricate designs address the limitations of existing techniques, allowing for precise object location and flexible background control.

3. The proposed pipeline handles real-world objects and complex backgrounds effectively.

4. The proposed zero-shot object customization is achieved without test-time optimization, allowing for control over the location, viewpoints, and background.

**Weaknesses:**

1. The proposed CustomNet focuses on the limitation of 256 × 256 resolution, which may affect the quality of generated images.

2. Non-rigid transformations and object style changes are not supported, limiting flexibility.

**Questions:**

1. How is CustomNet different from past methods for object customization? How does it achieve better identity preservation and varied output generation?

2. Can you explain the detailed designs mentioned in the text that allow for location and background control in CustomNet?

3. How does CustomNet handle real-world objects and complex backgrounds in its dataset construction pipeline, and what's the motivation?

4. Can you explain zero-shot object customization without test-time optimization and its multi-aspect control?

---

> ### Author Response · Authors · 2023-11-20
> **Official Response by Authors to Reviewer h7FN-- Part 1/2**
>
> ## **W1. Limitation of 256$\times$256 Resolution.**
>
>
>
> - **Limitation of Resolution**
>
>     - The constraint of the resolution of $256\times256$ can be addressed in two ways. For the first one, we can use an improved Zero1-to-3, *e.g.*, The Zero1-to-3 ++ [8], which has extended the resolution to $512\times512$. The second one is to train our CustomNet on large-resolution data from the current zero-1-to-3 checkpoint.
>
>
>     -  We update more real-world object customization cases in the update paper, shown in **Figures 12, 13, 14**.
>
> ***
>
> ## **W2. Non-rigid transformations and style changes.**
>
> - It can be solved by training our model on a larger web-scale dataset with our dataset construction pipeline. The Non-rigid transformations and style changes can be controlled by text prompts, which has been proven in the SOTA Text-to-Image Diffusion model. Current Text-to-image diffusion models present promising results by training with large web-scale (text, image) pair datasets. With our dataset construction pipeline, we can convert the web-scale text-image pair datasets into (text, image, and novel-view image) pair datasets, then train our model to achieve non-rigid transformations and style changes.
>
> ***
>
> ## **Q1. Difference between the proposed CustomNet and existing methods. How does CustomNet achieve better performance?**
>
>
> - **Difference from other works.**
>
>     - We should kindly remind you that we novelty incorporate the 3D novel view synthesis capability into the task of object customization, obtaining more flexible customization results with various backgrounds. Note that previous object customization methods (optimization-based Dreambooth[4], Textual Inversion[5], and encoder-based Paint-by-example[2], Anydoor[3]) do not consider the 3D properties of the inserted object but learn a compressed token embedding to represent the object visual information. Therefore, they struggle to balance identity preservation and output diversity.  For example, Paint-by-example[2] loses identity, while Anydoor [3] produces copy-paste results, lacking diversity. other optimization-based methods Dreambooth[4], and Textual Inversion[5] suffer long test-time optimization times and are prone to over-fitting.
>
>     - In 3D novel-view synthesis fields, following Zero-1-to-3, most works focus on how to improve it to generate better geometry and quality, e.g. Magic123 [6], One-2-3-45 [7]. They still have the limitation in Zero-1-to-3 that can not be applied in real-world customization applications directly. However, CustomNet provides such a possibility to bridge the 3D novel-view synthesis and object customization with flexible multiple controls.
>
> - **CustomNet achieves better identity preservation and varied output generation in the following three aspects**:
>
>     - **Model architecture.**  Other methods mainly extract object identity visual information into a compressed embedding vector (whose dimension is normally 1 $\times$ 768), Which makes it hard to preserve the identities of different objects. CustomNet preserves objects' identity by contacting objects corresponding latent with the UNet input latent, and sending them into UNet together. the object latent is the output of the VAE encoder, which can reconstruct the object input image by the VAE decoder. Therefore, the object latent contains the most information as the total model (VAE + Unet) can do.
>
>     - **Explicit control with 3D prior.** Other methods change the object generation by text control. The text prompt may be ambiguous in semantics, and the words that are not used be describe the object could also affect the object's identity. We design a separate viewpoints control branch that learns to warp the object into different view synthesis. 3D viewpoint warping is a rigid process that has less uncertainty than text description. Therefore, with our explicit control, we can preserve object identity better when generating various results.
>
>     - **Dataset construction.**  Other methods mainly use the text-image pair dataset for customization, while we further generate a multi-view for the object image which provides more information for the model to learn the object visual concepts.
>
> ***

---

> > ### Author Response · Authors · 2023-11-20
> > **Official Response by Authors to Reviewer h7FN-- Part 2/2**
> >
> > ## **Q2. Location control and background control.**
> >
> >
> > - **Location control** is achieved by specifying a bounding box where the object is located in the result-generated image. Assume that we are going to generate a customized image that the object will locate at [x, y, w, h], then we just need to resize the reference object image into the size of [w, h] and place its left-top corner at the [x, y] coordinate of a background-free image (this image is the same size as the target image being denoised but without background). We have demonstrated it in detail in the main paper, please see Sec. 3.1 and model pipeline Fig. 2.
> >
> > - **Background control**  is controlled by two branches: text generation and image composition. In text generation, the user can provide a text prompt that describes the background, and take it into text encoder input. In image composition, the user can provide a background image directly, and concat it with the UNet Input latent.
> >
> > ***
> >
> > ## **Q3. How to handle real-world images in dataset construction pipeline and motivation**
> >
> > - Given a real-world image, we need to process it to obtain the text description and object novel-view image as input.
> >
> >     - For text prompt, we send the input image into the BLIP2 model to generate the corresponding text description of the image
> >
> >     - For the novel-view images, we first use the BLIP-2 5T version to extract the foreground object with the instruction {"image": image, "prompt": "Question: What foreground objects are in the image? find them and separate them using commas. Answer:"}. Then we feed the object to SAM, SAM can receive text as input and output the segmentation mask of the corresponding object, and we can get the single object image by the segmentation mask. Finally, we use Zero-1-to-3 to generate novel-view images of the objects.
> >
> >
> >
> > - The motivation of dataset construction is to facilitate fusing real-world 3D prior in CustomNet training for better object identity preservation and harmonious natural generation. Training our model only with synthetic datasets will generate 'floating' artifacts between the object and background. Training with real-world data, the artifacts are alleviated significantly.
> >
> > ***
> >
> > ## **Q4.  Explanations of zero-shot object customization without test-time optimization and its multi-aspect control**
> >
> > - **Zero-shot object customization** is a task or scenario that which a user provides a reference object image that may not be seen in model training to do generation while preserving their identities.
> >
> > - **Test-time optimization** corresponds to the optimization-based methods used in customization. When users provide multiple images of the same object during inference, this method iterative optimizes model parameters to perform online training for these images for minutes to hours, and then the users can use the model to generate customized images. The phrase 'test-time' refers to the user inference process, and 'optimization' means the category of optimization method. CustomNet can directly perform customized generation when the user provides only a single image during inference. Therefore CustomNet can achieve generation **without test-time optimization**.
> >
> > - **Multi-aspect control** means that our CustomNet can control customized object generation by viewpoints, locations, text prompts, or background images simultaneously while other methods mainly control it only by text prompts.
> >
> >
> > ***
> >
> >
> > [1] Yukai Shi, et al. "TOSS: High-quality Text-guided Novel View Synthesis from a Single Image." ArXiv preprint arXiv:2310.10644.
> >
> > [2] Binxin Yang, et al. "Paint by Example: Exemplar-based Image Editing with Diffusion Models." In Proceedings of the IEEE/CVF Conference on Computer Vision and Pattern Recognition, 2023.
> >
> > [3] Xi Chen, et al. "AnyDoor: Zero-shot Object-level Image Customization." ArXiv preprint arXiv:2307.09481.
> >
> > [4] Nataniel Ruiz, et al. "DreamBooth: Fine Tuning Text-to-Image Diffusion Models for Subject-Driven Generation" In Proceedings of the IEEE/CVF Conference on Computer Vision and Pattern Recognition, 2023.
> >
> > [5] Rinon Gal, et al. "An Image is Worth One Word: Personalizing Text-to-Image Generation using Textual Inversion." In International Conference on Learning Representations, 2023.
> >
> > [6] Guocheng Qian, et al. "Magic123: One Image to High-Quality 3D Object
> > Generation Using Both 2D and 3D Diffusion Priors." ArXiv preprint arXiv:2306.17843.
> >
> >
> > [7] Minghua Liu, et al. "One-2-3-45: Any Single Image to 3D Mesh in 45
> > Seconds without Per-Shape Optimization." ArXiv preprint arXiv:2306.16928.
> >
> >
> > [8] Ruoxi Shi, et al. "Zero123++: a Single Image to Consistent Multi-view Diffusion Base Model." ArXiv preprint arXiv:2310.15110.

---

> ### Author Response · Authors · 2023-11-21
> **Sincerely Look Forward to Your Feedback**
>
> Dear Reviewer h7FN:
>
> Thanks again for all of your constructive comments and suggestions, which have helped us improve the quality and clarity of this paper
>
> We sincerely hope that our added experiments and analyses could address your concerns.
>
> Since the deadline for discussion is approaching, please feel free to let us know if there are any additional clarifications or experiments that we can offer. Your suggestions are highly appreciated.
>
> Best wishes,
>
> Authors

---

### Official Review · Reviewer_vmn1 · 2023-10-29

**Soundness:** 3 good
**Presentation:** 3 good
**Contribution:** 2 fair
**Rating:** 6
**Confidence:** 3

**Summary:**

The paper presents an method for inserting objects into scenes. The main contribution is a unified framework that allows users to specify the background, the object, and its location. The background can be specified by text description (generation) or actual image pixels (composition), while the object is specified by its background-subtracted image, the desired relative camera view, and its bounding box within the final image.

**Strengths:**

There are two main strengths of the paper:

1. The presented method is an end-to-end solution for the task of placing objects into scenes, which produces harmonious outputs easier than a multi-stage pipeline.

2. The presented results look convincing in both preserving the object's identity a and harmonizing the final composition.

**Weaknesses:**

The novelty of the approach is somewhat limited, as it essentially plugs background generation/compositing into the Zero-1-2 architecture. Placing the object into a localized bounding box as a guide instead of keeping it centralized would be straightforward to do in Zero-1-2.

An interesting component of the system is the synthetic dataset of objects composed onto backgrounds. Publishing that dataset for future research would add to the contribution.

**Questions:**

How come you need to specify both the camera translation and image location? Wouldn't it be enough to only encode camera rotation and the bounding box to place the object anywhere in the image?

You mention that Zero-1-to-3 method cannot produce non-centered objects. But they allow users to specify full 3D camera rotation and translation, which is a component that this paper inherits. Why wouldn't translation enable users to place objects off-center?

If I understood the section 3.2 correctly, to make Figure 2 more clear, you could show 2 branches as the input to the text encoder: "sitting on the beach" for the Generation branch and NULL for the Composition branch.  Although I may be wrong, since the model could possibly take in both  the background image and the textual description. What happens in that case in terms of the output?

Phrases like "delicate designs" and "intricate designs" seem out of place in this paper. The network design, if that's what this refers to, seems reasonable and straightforward.

It may be helpful to specify that R is a 3x3 matrix and T is a 3-vector. If that really is the case (as it is in Zero-1-2), then the rotation is not just a rotation, but also non-uniform scaling and sheer and possibly even reflection. Is that too many degrees of freedom to specify as input for a human user? Why not just specify the viewing direction with something like azimuth+altitude?

You did not define d in equation 2.

A figure visualizing some of the synthetic dataset images would be helpful.

Paper should discuss some remaining artifacts in the limitation section, such as:
  * Minor change of identity in Figure 1 Row 3 (look at the number 1 on the toy car)
  * Bad aspect ratio of the dog in Figure 1 Row 4 (it's too squished)
  * The white pixels around the character in Figure 4 Row 1 Last Column
Where do these come from and how would one go about improving them?

---

> ### Author Response · Authors · 2023-11-20
> **Official Response by Authors to Reviewer vmn1-- Part 1/3**
>
> ## **W1. Novelty and object placement.**
>
>
>
> ### **About Novelty**
> We would like to kindly remind you that CustomNet is designed for **the task of object customization** that can place the object into desired scenes harmoniously with the explicit viewpoint, location, and background controls. We are not a simple application of zero-1-to-3 in the object customization field, but we introduce it to image customization for the first time, by addressing the inherent limitations of zero-1-to-3.
> Compared to previous image customization methods, CustomNet is the *first to investigate the novel view synthesis in the customization field*.
> It can preserve the object's identity while controlling the viewpoint and increasing the output diversity.
>
> - **Improvements of zero-1-to-3.**
>
>     - Our module designs, together with the training dataset construction, are important improvements for the final outcome of object customization. They cannot be viewed as a simple addition to zero-1-to-3. They are specialized designs for object customization. Without those designs for object customization in the zero-1-to-3 model, the synthesized image lacks the background, and the object is centrally localized, which is far from our goal of object customization. The comparison to the Zero-1-to-3 limitation in our updated Figure 10 shows that our designs are necessary and effective.
>     Besides, the concurrent paper TOSS [1] also mentions in their paper that the extra text input could improve the generation quality of novel views over zero-1-to-3.
>     In our CustomNet, incorporating real-world data with text could also benefit the generation results, which is insightful to improve the novel-view synthesis model.
>
>
>     - Specifically, as described in the main paper, we demonstrate that Zero-1-to-3 has the following limitations when applied to object customization, and we make specific designs to address those limitations and adapt them to the task of object customization.
>
>         - I. Zero-1-to-3 cannot be controlled by text for background. **v.s.** We design a dual-attention mechanism to simultaneously control viewpoint and text.
>         - II. Zero-1-to-3 cannot generate/composite backgrounds with a given background image. **v.s.** We designed a unified pipeline to combine it.
>         - III. Zero-1-to-3 can only process images located in the image's center. **v.s.** We design the concatenation module to place the object in arbitrary locations flexibly.
>         - IV. Zero-1-to-3 usually generates unreal effects. **v.s.** We design a real dataset construction pipeline to extend its capability on real data.
>
> - **Difference from other works.**
>
>     - We should kindly remind you that we novelty incorporate the 3D novel view synthesis capability into the task of object customization, obtaining more flexible customization results with various backgrounds. Note that previous object customization methods (optimization-based Dreambooth[4], Textual Inversion[5], and encoder-based Paint-by-example[2], Anydoor[3]) do not consider the 3D properties of the inserted object but learn a compressed token embedding to represent the object visual information. Therefore, they struggle to balance identity preservation and output diversity.  For example, Paint-by-example[2] loses identity, while Anydoor [3] produces copy-paste results, lacking diversity. other optimization-based methods Dreambooth[4], and Textual Inversion[5] suffer long test-time optimization times and are prone to over-fitting.
>
>     - In 3D novel-view synthesis fields, following Zero-1-to-3, most works focus on how to improve it to generate better geometry and quality, e.g. Magic123 [6], One-2-3-45 [7]. They still have the limitation in Zero-1-to-3 that can not be applied in real-world customization applications directly. However, CustomNet provides such a possibility to bridge the 3D novel-view synthesis and object customization with flexible multiple controls.
>
>
> - **Object placement.** The original Zero-1-to-3 model struggles to synthesize novel views of an object at an arbitrary location, in their camera settings, they cannot control translation but only azimuth angle, polar angle, and size. Therefore, we introduce a localized bounding box as guidance.
> It is a simple but effective design for solving this problem.
> When applying the same location control strategy to zero-1-to-3 directly, we can see that significant artifacts would occur, while our CustomNet has reasonable results in the updated **Figure 10**.
>
>
> ***
>
>
> ## **W2. Dataset construction pipeline.**
>
> - Thank you for your advice. Our dataset can help CustomNet obtain more harmonious results since the real images serve as the model target. We will release the
> dataset, together with the full pipeline for constructing the dataset to contribute to the development of future research.
>
>
> ***

---

> ### Author Response · Authors · 2023-11-20
> **Official Response by Authors to Reviewer vmn1-- Part 2/3**
>
> ## **Q1. Camera translation and object location.**
>
> - We should clarify that the definition and design of **RT** is consistent with the setting in Zero-1-to-3. More specifically, Zero-1-to-3 renders 3D objects into images with a predefined camera pose that always points at the center of the object.
> This means the viewpoint is actually controlled by polar angle $\theta$, azimuth angle $\phi$, and radius $r$ (distance away from the center), not the full rotation and translation matrices, resulting all objects to be central of the rendered image. Thus, the Zero-1-to-3 can only synthesize objects at the image center, lacking the capacity to place the object at the **arbitrary spatial location** in the image.
>
> - Note that zero-1-to-3 cannot perform translation and can only generate objects localized centrally. We tackle this problem by specifying the bounding box at the desired location to the latent. This design is actually consistent with your comments.
>
>
> ***
>
>
> ## **Q2. Translation in Zero-1-to-3.**
>
> - Note that the zero-1-to-3 model cannot synthesize objects with arbitrary spatial translations, and can only control the size of the objects. Please refer to Q1 for the specified reasons. We address this limitation by placing objects in a specified location bounding box.
>
>
> ***
>
>
> ## **Q3. Generation and composition branches.**
>
> - You are right. We provide two ways to control the background: textual description and given background image, referred to as generation and composition branches, respectively.
> For the generation branch, We added a new text branch to enable generating the background complying with the given text description with the Dual-Attention module.
>
> - As for the composition branch, when both given text and background image, the model generation will be dominated by the background images, as it provided more information than text when training.
>
>
> ***
>
> ## **Q4. Delicate or intricate.**
>
> - We will modify our phrase more precisely. Our goal that customizing an object with identity preservation and diverse control is straightforward. However, to achieve this goal, we have several careful designs from dataset construction (both synthetic and real datasets) to model architecture (DualAttn for text and viewpoints, and location control in initial latent concat).
>
> ***
>
>
> ## **Q5. Definition of RT.**
>
> - The **RT** is controled by the polar angle $\theta$, azimuth angle $\phi$, and radius $r$ (distance away from the center). Following your advice, We added the detailed statements in the paper in **Appendix A.2**, which is the same as the following sentence. "The definition and design of RT is consistent with the setting in Zero-1-to-3. In Zero-1-to-3, they render 3D objects into images with a predefined camera pose that always points at the center of the object. This means the viewpoint is actually controlled by polar angle $\theta$, azimuth angle $\phi$, and radius $r$ (distance away from the center),  resulting in all objects being central to the rendered image. Therefore, the rotation matrix R can only control polar angle $\theta$, azimuth angle $\phi$, and the translation vector T can only control the radius $r$ between the object and camera. Thus the Zero-1-to-3 can only synthesize objects at the image center, lacking the capacity to place the object at the arbitrary location in the image. We tackle this problem by specifying the bounding box at the desired location to the latent."
>
> ***
>
>
> ## **Q6. Definition of d.**
>
> - Thank you for pointing out the mistakes. We have updated the definition on it in the **Sec. 3.2**.

---

> > ### Comment · Reviewer_vmn1 · 2023-11-20
> > **Re: Q1, Q5**
> >
> > Thank you for the explanation. I have a follow-up question:
> >
> > Is the radius parameter needed now that you can specify a whole bounding box? What does it achieve? Does it further control the scale of the object within the bounding box? Does it control the camera field-of-view (a larger radius may yield an almost orthographic projection)? Is it largely ignored and the bounding box overpowers the radius parameter?

---

> > > ### Author Response · Authors · 2023-11-21
> > > **Reply to Reviewer vmn1: Re: Q1, Q5**
> > >
> > > We deeply appreciate the time you have taken to review our responses. Regarding the radius parameters, we would like to clarify that we donot specify them as the bounding box has a more powerful ability to control both the scale and location of the object. We are going to further discuss it in the followings.
> > >
> > > - In the Zero-1-to-3 dataset, objects are rendered with a radius sampled from a certain range to control the centred object scale. However, in practical testing, the control ability of the radius parameter is not prominent due to the sampling strategy used in their dataset (the range is a little narrow).
> > >
> > > - In our CustomNet, to better handle the real world scenario, we opt to use the bounding box control both the scale and location of the object instead of the radius parameters. In our settings, we set the radius as default value as it is inherited from the Zero-1-to-3.

---

> > > > ### Comment · Reviewer_vmn1 · 2023-11-21
> > > > **Radius parameter**
> > > >
> > > > You may want to mention in the paper that you set the radius to some default value (is it 1.0?) and use only the rotation parameters of the camera view and the bounding box.
> > > >
> > > > This raises another follow up question: what happens if the bounding box is squished (e.g. very wide and short)? Does the object become non-uniformly scaled (appears too wide)? Or does it just scale down to the smaller dimension and still retain the expected aspect ratio?

---

> > > > > ### Author Response · Authors · 2023-11-21
> > > > > **Reply to Reviewer vmn1: Radius parameter**
> > > > >
> > > > > Thank you for your advice, we will update the settings of the radius parameters in paper.
> > > > >
> > > > >
> > > > > The bounding box will affect the aspect ratio of the object. If the bounding box is squished, the object will look wide.

---

> ### Author Response · Authors · 2023-11-20
> **Official Response by Authors to Reviewer vmn1-- Part 3/3**
>
> ## **Q7. Visualization samples of the dataset.**
>
> - We have updated more dataset visualization samples in the paper, please see the updated **Figure 9**.
>
> ***
>
> ## **Q8. Artifacts in the limitation section.**
>
> - First, this is a hard case in which other models usually fail to preserve their identity, see the updated **Figure 11**. Some minor changes would indeed occur when customizing objects with highly detailed textures. Compared to previous methods, our identity preservation is highly competitive.
> We will mention it in the limitation, and continuously improve the identity in future work.
>
> - The white square in the background causes the artifacts. When testing another method, we place a white square to mask the original object.
> When testing our CustomNet, we mistakenly keep the white square and the model interprets the white pixel as the background.
> Our CustomNet will generate a good image without artifacts when provided with a pure background image. Please refer to the revised **Figure 4** for further clarification.
>
> - The model allows for modifications to be made to the location bounding box and viewpoints as specified by the user. We have set an overhead view of the dog in the figure, resulting in it appearing squished.
> Such an artifact can be addressed by modifying the location bounding box and viewpoints.
>
>
> ***
>
> [1] Yukai Shi, et al. "TOSS: High-quality Text-guided Novel View Synthesis from a Single Image." ArXiv preprint arXiv:2310.10644.
>
> [2] Binxin Yang, et al. "Paint by Example: Exemplar-based Image Editing with Diffusion Models." In Proceedings of the IEEE/CVF Conference on Computer Vision and Pattern Recognition, 2023.
>
> [3] Xi Chen, et al. "AnyDoor: Zero-shot Object-level Image Customization." ArXiv preprint arXiv:2307.09481.
>
> [4] Nataniel Ruiz, et al. "DreamBooth: Fine Tuning Text-to-Image Diffusion Models for Subject-Driven Generation" In Proceedings of the IEEE/CVF Conference on Computer Vision and Pattern Recognition, 2023.
>
> [5] Rinon Gal, et al. "An Image is Worth One Word: Personalizing Text-to-Image Generation using Textual Inversion." In International Conference on Learning Representations, 2023.
>
> [6] Guocheng Qian, et al. "Magic123: One Image to High-Quality 3D Object
> Generation Using Both 2D and 3D Diffusion Priors." ArXiv preprint arXiv:2306.17843.
>
>
> [7] Minghua Liu, et al. "One-2-3-45: Any Single Image to 3D Mesh in 45
> Seconds without Per-Shape Optimization." ArXiv preprint arXiv:2306.16928.

---

> ### Comment · Reviewer_vmn1 · 2023-11-20
> **Raising the rating**
>
> Thanks to the authors for the rebuttal. In light of all the clarifications, additional figures, and open-sourcing the synthetic dataset+pipeline, I am raising my rating.

---

> > ### Author Response · Authors · 2023-11-21
> > **Reply to Reviewer vmn1:  Raising the rating**
> >
> > We express our sincere gratitude to you for taking the time to review our paper and for the increased score. Your valuable feedback has been instrumental in improving our work, and we are delighted that our response has been helpful.

---

### Official Review · Reviewer_96bJ · 2023-11-01

**Soundness:** 3 good
**Presentation:** 3 good
**Contribution:** 2 fair
**Rating:** 5
**Confidence:** 4

**Summary:**

The paper presents a novel approach to text-to-image generation that integrates 3D novel view synthesis for enhanced object customization. Unlike traditional methods that have challenges with identity preservation and limited customization, CustomNet offers explicit viewpoint control, location adjustments, and flexible background generation. It leverages a new dataset construction pipeline using both synthetic multiview data and natural images. The result is a model that achieves zero-shot object customization with improved identity preservation, diverse viewpoints, and harmonious outputs.

**Strengths:**

1. The paper is well-written and easy to follow.
2. The authors present a comprehensive solution that seamlessly integrates background inpainting into the Zero-1-to-3 pipeline, addressing its potential limitation in altering the background or location of the object.
3. The paper utilizes the latest vision foundation models, such as SAM and BLIP, in conjunction with "Zero-1-to-3," to construct a dataset that supports the training of the proposed unified framework. The inverse data generation pipeline which decompose the netural image into the training component is interesting.

**Weaknesses:**

1. A significant concern with this paper is its limited novelty. It leans heavily on the "Zero-1-to-3" model as its foundation. While the addition of background inpainting offers an enhancement, the core mechanism—explicit 3D novel view synthesis—remains unchanged. Moreover, it inherits constraints from "Zero-1-to-3", particularly the resolution limitation of 256 × 256, which might not be practical for real-world scenarios.
2. The approach to decompose foreground objects using SAM and augmenting them using "Zero-1-to-3" is intriguing. However, there are lingering questions about the methodology. For instance, considering that SAM segments every element in a scene, how does the paper specifically determine the foreground object? Furthermore, if the selected foreground object falls outside the "Zero-1-to-3" training domain, is the efficacy of the method compromised?
3. The paper mostly compares CustomNet with "Zero-1-to-3" and a few other models. A broader discussion with state-of-the-art models in the domain would have given a better quality assessment of CustomNet's performance. For example, [1] also employs stable diffusion for background inpainting while preserving the appearance of foreground objects through a dual-branch composition.

[1] Li, Siyuan, et al. "OVTrack: Open-Vocabulary Multiple Object Tracking." Proceedings of the IEEE/CVF Conference on Computer Vision and Pattern Recognition. 2023.

**Questions:**

Please see above weakness.

---

> ### Author Response · Authors · 2023-11-20
> **Official Response by Authors to Reviewer 96bJ -- Part 1/2**
>
> ## **W1. Novelty and Resolution Constraint.**
>
>
> ### **About Novelty**
> We would like to kindly remind that CustomNet is designed for **the task of object customization** that can place the object into desired scenes harmoniously with the explicit viewpoint, location, and background controls. We are not a simple application of zero-1-to-3 in the object customization field, but we introduce it to image customization for the first time, by addressing the inherent limitations of zero-1-to-3.
> Compared to previous image customization methods, CustomNet is the *first to investigate the novel view synthesis in the customization field*.
> It can preserve the object's identity while controlling the viewpoint and increasing the output diversity.
>
> - **Improvements of zero-1-to-3.**
>
>     - Our module designs, together with the training dataset construction, are important improvements for the final outcome of object customization. They cannot be viewed as a simple addition to zero-1-to-3. They are specialized designs for object customization. Without those designs for object customization in the zero-1-to-3 model, the synthesized image lacks the background, and the object is centrally localized, which is far from our goal of object customization. The comparison to the Zero-1-to-3 limitation in our updated Figure 10 shows that our designs are necessary and effective.
>     Besides, the concurrent paper TOSS [1] also mentions in their paper that the extra text input could improve the generation quality of novel views over zero-1-to-3.
>     In our CustomNet, incorporating real-world data with text could also benefit the generation results, which is insightful to improve the novel-view synthesis model.
>
>
>     - Specifically, as described in the main paper, we demonstrate that Zero-1-to-3 has the following limitations when applied in object customization, and we make specific designs to address those limitations and adapt them to the task of object customization.
>
>         - I. Zero-1-to-3 cannot be controlled by text for background. **v.s.** We design a dual-attention mechanism to simultaneously control viewpoint and text.
>         - II. Zero-1-to-3 cannot generate/composite backgrounds with a given background image. **v.s.** We designed a unified pipeline to combine it.
>         - III. Zero-1-to-3 can only process images located in the image's center. **v.s.** We design the concatenation module to place the object in arbitrary locations flexibly.
>         - IV. Zero-1-to-3 usually generates unreal effects. **v.s.** We design a real dataset construction pipeline to extend its capability on real data.
>
> - **Difference from other works.**
>
>     - We should kindly remind that we novelty incorporate the 3D novel view synthesis capability into the task of object customization, obtaining more flexible customization results with various backgrounds. Note that previous object customization methods (optimization-based Dreambooth[4], Textual Inversion[5], and encoder-based Paint-by-example[2], Anydoor[3]) do not consider the 3D properties of the inserted object but learn a compressed token embedding to represent the object visual information. Therefore, they struggle to balance identity preservation and output diversity.  For example, Paint-by-example[2] loses identity, while Anydoor [3] produces copy-paste results, lacking diversity. Other optimization-based methods Dreambooth[4], and Textual Inversion[5] suffer long test-time optimization times and are prone to over-fitting.
>
>     - In 3D novel-view synthesis fields, following Zero-1-to-3, most works focus on how to improve it to generate better geometry and quality, e.g. Magic123 [6], One-2-3-45 [7]. They still have the limitation in Zero-1-to-3 that can not be applied in real-world customization applications directly. However, CustomNet provides such a possibility to bridge the 3D novel-view synthesis and object customization with flexible multiple controls.
>
> ### **Limitation of Resolution**
>
> - The constraint of the resolution of $256\times256$ can be addressed by in two ways. For the first one, we can use an improved Zero1-to-3, *e.g.*, The Zero1-to-3 ++ [8], which has extended the resolution to $512\times512$. The second one is to train our CustomNet on large-resolution data from the current zero-1-to-3 checkpoint.
>
> -  We update more real-world object customization cases in the update paper, shown in **Figure 12, 13, 14**.

---

> ### Author Response · Authors · 2023-11-20
> **Official Response by Authors to Reviewer 96bJ -- Part 2/2**
>
> ## **W2. Details of the Dataset construction pipeline.**
>
> We provide more details about our dataset construction pipeline.
>
> - How to specifically determine the foreground object:  Given an image, We use BLIP2 (Github repo: https://github.com/salesforce/LAVIS/blob/main/examples/blip2_instructed_generation.ipynb) to extract the foreground object with the following instruction:
> '''
> {"image": image, "prompt": "Question: What foreground objects are in the image? find them and separate them using commas. Answer:"}
> '''
> Then we feed the queried object and its corresponding text to SAM, SAM can receive text as input and output the segmentation mask of the corresponding object.
>
> - When the selected foreground object falls outside the Zero-1-to-3 training domain, we may get unsatisfying results. We have tried to alleviate this issue in our CustomNet
>     1. We jointly train with both synthetic and real datasets, which will complement each other and narrow the gap between the synthetic and real datasets.
>     2. During training, though the input object image may be unsatisfying, we adopt the original natural image with the object as the target. It is helpful for model training as the target image provides information about how to place an object into a result image harmoniously and naturally, which helps in real-world scenarios. On the other hand, the synthetic dataset endows the model with the warping ability among multiple views. Combined with the two datasets, CustomNet demonstrates natural and high-fidelity customization.
>     3. In our CustomeNet, except for the image input, CustomNet also receives text as input. The text helps to improve the viewpoints control ability that Zero-1-to-3 is hard to manage, as shown in the updated **Figure 10**.
>
>
> ***
>
>
> ## **W3. Broad Discussion with other methods.**
>
> - To achieve customization, We can use a one-stage model like the optimization-based method Dreambooth[4] or Encoder-based Paint-by-example [2]. The optimization-based method suffers long test-time optimization times and is prone to over-fitting, while the encoder-based method may lose identity. We can also use two-stage methods that first get the foreground object image, then paste it to a new background directly or with diffusion inpainting. The OVTrack [9] uses stable diffusion background inpainting while preserving the appearance of foreground objects for a data hallucination strategy tailored to appearance modeling in multi-object tracking. Our Customnet can generate different views of the foreground object, and use text to generate background for it.
>
> - We have conducted quantitative and qualitative comparisons to the state-of-the-art encoder-based and optimization-based image customization methods in Sec. 4 of the main paper. We update further discussion with the OVTrack[9] and other models, such as Blended latent diffusion[10], etc. in **the Discussion in Sec. 3 in the updated paper**.
>
>
>
> ***
>
>
> [1] Yukai Shi, et al. "TOSS: High-quality Text-guided Novel View Synthesis from a Single Image." ArXiv preprint arXiv:2310.10644.
>
> [2] Binxin Yang, et al. "Paint by Example: Exemplar-based Image Editing with Diffusion Models." In Proceedings of the IEEE/CVF Conference on Computer Vision and Pattern Recognition, 2023.
>
> [3] Xi Chen, et al. "AnyDoor: Zero-shot Object-level Image Customization." ArXiv preprint arXiv:2307.09481.
>
> [4] Nataniel Ruiz, et al. "DreamBooth: Fine Tuning Text-to-Image Diffusion Models for Subject-Driven Generation" In Proceedings of the IEEE/CVF Conference on Computer Vision and Pattern Recognition, 2023.
>
> [5] Rinon Gal, et al. "An Image is Worth One Word: Personalizing Text-to-Image Generation using Textual Inversion." In International Conference on Learning Representations, 2023.
>
> [6] Guocheng Qian, et al. "Magic123: One Image to High-Quality 3D Object
> Generation Using Both 2D and 3D Diffusion Priors." ArXiv preprint arXiv:2306.17843.
>
>
> [7] Minghua Liu, et al. "One-2-3-45: Any Single Image to 3D Mesh in 45
> Seconds without Per-Shape Optimization." ArXiv preprint arXiv:2306.16928.
>
>
> [8] Ruoxi Shi, et al. "Zero123++: a Single Image to Consistent Multi-view Diffusion Base Model." ArXiv preprint arXiv:2310.15110.
>
>
> [9] Li, Siyuan, et al. "OVTrack: Open-Vocabulary Multiple Object Tracking." Proceedings of the IEEE/CVF Conference on Computer Vision and Pattern Recognition. 2023.
>
> [10] Omri Avrahami, et al. "Blended Latent Diffusion." In SIGGRAPH, 2023.

---

> ### Author Response · Authors · 2023-11-21
> **Sincerely Look Forward to Your Feedback**
>
> Dear Reviewer 96bJ:
>
> Thanks again for all of your constructive comments and suggestions, which have helped us improve the quality and clarity of this paper
>
> We sincerely hope that our added experiments and analyses could address your concerns.
>
> Since the deadline for discussion is approaching, please feel free to let us know if there are any additional clarifications or experiments that we can offer. Your suggestions are highly appreciated.
>
> Best wishes,
>
> Authors

---

### Author Response · Authors · 2023-11-20
**General response to all reviewers**

We sincerely appreciate all reviewers' efforts in reviewing our paper and giving insightful comments and valuable suggestions. **We recommend reviewing our modification of both text and new figures in the paper. We update some new discussions in A.1, A.2, A.3, and update new figures in Fig. 9, 10, 11, 12, 13, 14**

In the following, We are going to summarize the strengths of our work which reviewers generally acknowledge.


- **Unified Framework.** CustomNet is a unified object customization framework that can generate harmonious customized images with explicit viewpoint, location, and background controls simultaneously, while ensuring identity preservation. It is an end-to-end solution for placing an object into desired scenes with the desired viewpoint, which simplifies current complex multi-stage pipelines.

- **Data Construction.**
The integration of the latest comprehension models, such as SAM and BLIP2, into our proposed dataset construction pipeline is interesting. The proposed pipeline effectively handles real-world objects and complex backgrounds.

- **Performance.** The results of CustomNet effectively demonstrate the preservation of the object's identity and harmony in the final composition. The zero-shot object customization achieved without test-time optimization is a significant advancement.

---

### Author Response · Authors · 2023-11-23
**A kind reminder regarding our response**

Dear reviewers,

Thank you for dedicating your time and effort in reviewing our paper. We have responded to all comments in the rebuttal. If you have any additional comments or questions, please feel free to let us know. As the rebuttal period is approaching its end, we would like to express our gratitude for your attention to our submission.

Best regards,

The Authors

---

### Meta-Review · Area_Chair_uXg6 · 2023-12-04

**Metareview:**

This work extends the object cutomized image generation by incorporating 3D view synthesis. To achieve this, they build their model on the zero-1-to-3 and design module and dataset for background generation. The experiment demonstrates the high quality of the generated cutomized object images.

**Justification For Why Not Higher Score:**

- The overall framework is largely built on the zero-1-to-3 model, with some modules introduced for generating background images. The novelty of this work is limited.

- The reviewers share the similar concerns on the significance of the technical contribution, considering the existence of zero-1-to-3. But the authors do not provide convincing rebuttal to this core questions.

- The reviewers raised some detailed questions, such as (1) why the rotation and translation cannot be unified represented; (2) why zero-1-to-3 cannot perform object translation. The authors did not address them well.

Overall, the idea proposed by this work is interesting. But the work is largely built up on zero-1-to-3 and also inherits its limiations. It does not make essential progress along this direction.

**Justification For Why Not Lower Score:**

NA

---

### Decision · Program_Chairs · 2024-01-16

Reject